# Personalized Stem Cell-Based Regeneration in Spinal Cord Injury Care

**DOI:** 10.3390/ijms26083874

**Published:** 2025-04-19

**Authors:** Sasi Kumar Jagadeesan, Ryan Vimukthie Sandarage, Sathya Mathiyalagan, Eve Chung Tsai

**Affiliations:** 1Department of Neurosciences, Faculty of Medicine, University of Ottawa, Ottawa, ON K1H 8M5, Canada; sjagadeesan@ohri.ca (S.K.J.); rsandarage@toh.ca (R.V.S.); 2Neuroscience Program, Ottawa Hospital Research Institute, The Ottawa Hospital, Ottawa, ON K1H 8L6, Canada; 3Division of Neurosurgery, Department of Surgery, The Ottawa Hospital, Ottawa, ON K1H 8L6, Canada; 4Faculty of Health Sciences, University of Ottawa, Ottawa, ON K1N 6N5, Canada; smath112@uottawa.ca

**Keywords:** spinal cord injury, adult human spinal cord, personalized medicine, induced pluripotent stem cells, neural/stem progenitor cells, neuroregeneration, axonal regrowth, functional recovery, transcriptomics

## Abstract

Spinal cord injury (SCI) remains a major clinical challenge, with limited therapeutic options for restoring lost neurological function. While efforts to mitigate secondary damage have improved early-phase management, achieving sustained neurorepair and functional recovery remains elusive. Advances in stem cell engineering and regenerative medicine have opened new avenues for targeted interventions, particularly through the transplantation of neural stem/progenitor cells (NSPCs), induced pluripotent stem cells (iPSCs), and mesenchymal stem cells (MSCs). However, patient-specific factors such as cellular senescence, genetic and epigenetic variability, injury microenvironment, and comorbidities influence the efficacy of stem cell therapies by affecting graft survival and differentiation. Overcoming these challenges necessitates cutting-edge technologies, including single-cell transcriptomics, CRISPR-mediated hypoimmunogenic engineering, and biomaterial-based delivery platforms, which enable personalized and precision-driven SCI repair. Leveraging these advancements may help stem cell therapies overcome translational barriers and establish clinically viable regenerative solutions. This review explores the intersection of patient-specific variability, bioengineering innovations, and transcriptomic-guided precision medicine to define the next frontier in SCI therapy.

## 1. Introduction

Spinal cord injury (SCI) results in irreversible motor, sensory, and autonomic dysfunction, presenting a significant challenge for regenerative medicine [1,2]. Despite decades of research, no curative treatment exists, and current clinical strategies focus primarily on stabilizing the injury site, preventing secondary damage, and maximizing residual function [3,4,5]. Surgical decompression, pharmacological interventions, and rehabilitative therapies provide some neuroprotection but fail to restore lost neural circuits [6,7]. Advanced assistive technologies, such as epidural electrical stimulation (EES) and robotic exoskeletons, have demonstrated promise in improving motor function but remain limited by high variability in efficacy, cost constraints, and their inability to regenerate damaged neural tissue [8,9]. Given these limitations, the future of SCI treatment hinges on regenerative strategies that integrate cell replacement, immune modulation, and biomaterial-based microenvironmental engineering to facilitate functional recovery and neural repair.

Stem cell-based therapies offer a compelling approach for repopulating lost neural populations, restoring connectivity, and modulating the post-injury microenvironment [10,11]. Neural stem/progenitor cells (NSPCs), induced pluripotent stem cells (iPSCs), and mesenchymal stem cells (MSCs) have shown promise in remyelination, synaptogenesis, and neuroprotection [11,12]. However, patient-specific variability including age-related cellular senescence, chronic inflammation, genetic and epigenetic diversity, injury severity, and immune response dynamics profoundly influences therapeutic outcomes [12,13,14,15]. These factors could directly impact graft survival, differentiation efficiency, and host integration, highlighting the need to move beyond standardized transplantation protocols. Advances in single-cell transcriptomics, spatial mapping, and CRISPR-based immune engineering now enable a systems-level approach to SCI repair, in which therapies could be tailored to individual molecular landscapes to optimize cellular engraftment and neuroregeneration [16,17]. Clinical studies have highlighted the need for patient-specific approaches in SCI therapy [12,15]. A phase I/II clinical trial evaluating human spinal cord-derived NSPCs in patients with thoracic SCI demonstrated significant variability in cell survival and integration, highlighting the influence of injury-specific microenvironments on graft efficacy [18]. Another preclinical study showed that transplantation of region-specific iPSC-derived neural progenitors enhanced synaptic integration and functional connectivity in a rodent model of SCI, emphasizing the need for lineage-optimized differentiation strategies [19]. Bioengineering innovations have significantly expanded the therapeutic potential of stem cell-based interventions by addressing key structural and functional challenges in neural repair [20,21]. Specifically, hydrogel-based scaffolds, nanofiber-aligned matrices, and 3D-bioprinted constructs provide essential structural support, enhance cellular retention and modulate the inflammatory milieu. Moreover, functionalized hydrogels infused with brain-derived neurotrophic factor (BDNF) and vascular endothelial growth factor (VEGF) promote angiogenesis, synaptic plasticity, and neuronal survival, while scaffolds delivering chondroitinase ABC (ChABC) enzymatically degrade glial scar components, thereby facilitating axonal extension and synaptic reconnection [22,23,24]. Additionally, electrically conductive biomaterials hold promise for restoring electrophysiological function by enhancing neural activity and supporting synaptic remodeling, further advancing the integration of bioengineered platforms in precision regenerative medicine [20].

Despite these advancements, the hostile post-injury environment continues to limit functional recovery, with molecular inhibitors impeding axonal regrowth and synaptic reconnection. Emerging interventions aimed at neutralizing these inhibitory cues have demonstrated promise in preclinical models, highlighting the need for targeted strategies to enhance neuroplasticity and circuit reformation [25,26]. Concurrently, hypoimmunogenic stem cell engineering, biomaterial-assisted immune modulation, and patient-specific immune profiling are being developed to reduce graft rejection and enhance the long-term survival of transplanted cells [27]. A multimodal approach, integrating stem cell transplantation, bioengineered scaffolds, adaptive neurostimulation, and immune modulation, offers a promising path toward clinically viable, precision-guided therapies. Machine learning-based transcriptomic analyses now enable patient stratification, identifying individuals most likely to benefit from specific regenerative interventions [28,29]. Additionally, single-cell RNA sequencing (scRNA-seq) and spatial transcriptomics provide a high-resolution molecular blueprint of injury-specific dynamics, allowing for tailored stem cell differentiation and biomaterial customization [17,30]. These developments collectively mark a paradigm shift from conventional one-size-fits-all strategies to personalized regenerative medicine approaches that align biomaterials, transcriptomics, and immune engineering with patient-specific injury profiles. This review explores the intersection of bioengineering, patient-specific variability, and transcriptomics-driven precision medicine in advancing SCI therapeutics. By integrating these cutting-edge approaches, stem cell-based therapies may overcome translational barriers and drive next-generation solutions for restoring neurological function and improving long-term patient outcomes.

## 2. Stem Cell Therapy for SCI

While conventional interventions primarily focus on stabilizing the injury site and mitigating secondary damage, stem cell-based therapies aim to reconstruct neural circuits, promote remyelination, and modulate the inflammatory microenvironment to enhance functional recovery [10,14]. However, the success of these therapies is contingent on several key factors, including graft survival, differentiation potential, and host integration, all of which are dictated by injury-specific and patient-specific variables (Figure 1) [11,12]. Among the most promising stem cell candidates, NSPCs, iPSCs, and MSCs exhibit distinct but complementary regenerative properties. NSPCs integrate into host spinal circuits, differentiating into neurons, oligodendrocytes, and astrocytes, thereby supporting axonal regeneration and remyelination [31,32]. Their secretion of neurotrophic factors, including brain-derived neurotrophic factor (BDNF) and glial cell-derived neurotrophic factor (GDNF), activates MAPK/ERK and PI3K/Akt signaling pathways, which are critical for neuronal survival, axonal growth, and synaptic plasticity [32,33]. Despite their potential, NSPCs face substantial challenges in transplantation, particularly survival, immune evasion, and efficient integration into the inflammatory injury niche [19,34]. Preclinical studies have demonstrated that hydrogel-based encapsulation enhances NSPC survival by protecting cells from oxidative stress and immune-mediated apoptosis, leading to improved graft retention and functional recovery [35,36]. Similarly, scaffold-supported cell delivery has shown promise in improving NSPC engraftment by mimicking the extracellular matrix and providing biochemical cues that enhance differentiation [24]. On the other hand, iPSC-derived neural progenitors offer a scalable and patient-specific alternative, capable of differentiating into diverse neural and glial lineages by regulating Wnt/β-catenin, Sonic Hedgehog (SHH), and Notch signaling pathways [37,38]. SHH gradients are particularly critical for oligodendrocyte specification, while Wnt signaling influences neuronal differentiation [39]. Despite their therapeutic potential, iPSCs are challenged by tumorigenic risks, susceptibility to oxidative stress, and differentiation inconsistencies, necessitating bioengineered scaffolds and small-molecule modulators to enhance survival, functional maturation, and long-term stability [40,41]. A phase I clinical trial using iPSC-derived neural progenitors in SCI patients demonstrated partial sensory and motor improvements, although variability in integration remained a critical challenge [42].

Alternatively, MSCs function primarily through paracrine signaling, exerting immunomodulatory and neuroprotective effects rather than direct neuronal replacement [43]. By secreting anti-inflammatory cytokines such as interleukin-10 (IL-10) and transforming growth factor-beta (TGF-β), MSCs polarize microglia from a pro-inflammatory (M1) to an anti-inflammatory (M2) phenotype, thereby attenuating neuroinflammation and facilitating tissue repair [44]. Furthermore, MSC-derived vascular endothelial growth factor (VEGF) enhances angiogenesis and vascular remodeling, promoting oxygen and nutrient delivery to injured tissue [45]. A clinical study analyzing MSC-based therapies for SCI highlighted the variability in clinical outcomes due to differences in cell sources, administration routes, and patient selection criteria, emphasizing the need for standardized protocols to enhance therapeutic efficacy [43,46]. Despite significant advances in stem cell transplantation, the inhibitory microenvironment of the injured spinal cord remains a major barrier to functional recovery [7,47]. Myelin-associated inhibitory molecules, including Nogo, myelin-associated glycoprotein (MAG), and leucine-rich repeat and immunoglobulin domain-containing Nogo receptor-interacting protein-1 (LINGO-1), activate the RhoA/ROCK signaling cascade, leading to growth cone collapse and impaired axonal regeneration [25]. Strategies targeting these molecular inhibitors, such as anti-Nogo antibodies, LINGO-1 antagonists, and MAG inhibitors, have shown promise in neutralizing inhibitory cues, enabling neurite outgrowth and enhancing axonal plasticity [48,49]. Collectively, these stem cell-based approaches offer promising avenues for neuroprotection and regeneration; however, overcoming the inhibitory microenvironment remains a critical challenge for achieving sustained functional recovery.

## 3. Advancements in Bioengineering for Stem Cell-Based SCI Repair

Bioengineering strategies have been developed to enhance the efficacy of stem cell-based therapies by overcoming the inhibitory microenvironment that impedes functional recovery, improving graft survival, optimizing neural integration, and facilitating axonal regeneration in SCI [20,21,36]. Emerging technologies, including biomaterial scaffolds, hydrogel-based delivery platforms, and 3D bioprinting, provide critical tools to optimize stem cell viability, differentiation, and host-circuit integration, while neurotrophic factor-infused hydrogels, enriched with BDNF and VEGF, create a biomimetic niche that enhances neuronal survival, synaptic remodeling, glial support, and stem cell retention (Figure 2) [50,51]. In parallel, the enzymatic modulation of chondroitin sulfate proteoglycans (CSPGs) via chondroitinase ABC (ChABC) continues to show promise in overcoming glial scar-associated barriers, thereby facilitating axonal regeneration [23]. Preclinical studies have demonstrated that aligned nanofiber scaffolds promote axonal regeneration and synaptic reconnection in SCI models, improving functional recovery [50,52]. In a canine L2 SCI model, an aligned fibrin nanofiber hydrogel provided a structured fiber bridge that supported cellular adhesion, facilitated directional axonal regrowth, and successfully reconnected nerve fibers between the rostral and caudal stumps, ultimately enhancing motor function recovery as confirmed by diffusion tensor imaging [53]. To further optimize therapeutic outcomes, electrically conductive biomaterials, including graphene and conductive polymers, have been explored as a means of restoring electrophysiological connectivity and facilitating neural network reactivation [20]. In parallel, 3D bioprinting technologies have revolutionized scaffold fabrication, enabling the construction of patient-specific biomimetic constructs that integrate regionally defined extracellular matrix components, growth factors, and stem cells in precisely controlled spatial arrangements [36]. A recent study using 3D-bioprinted hydrogel scaffolds seeded with iPSC-derived neural progenitors showed enhanced graft survival and axonal extension, demonstrating their potential for personalized regenerative therapies [54]. Additionally, iPSC-derived spinal cord organoids have provided a preclinical model for optimizing patient-matched regenerative strategies. These organoid models faithfully replicated the cytoarchitecture and cellular heterogeneity of the spinal cord, allowing for improved prediction of graft behavior and functional integration [55,56]. Alongside molecular and biomaterial-based interventions, electrical stimulation modalities, including epidural electrical stimulation (EES) and functional electrical stimulation (FES), have emerged as powerful adjuncts to stem cell-based therapies, facilitating circuit reactivation and synaptic plasticity enhancement [8]. EES has been shown to increase the excitability of residual supraspinal pathways, improving voluntary motor function, while FES directly stimulates paralyzed muscle groups, reinforcing neuroplasticity and synaptic remodeling [57]. A clinical study demonstrated that EES in combination with intensive rehabilitation resulted in significant motor recovery in individuals with chronic SCI, reinforcing the potential for neurostimulation-based approaches in regenerative strategies [58]. Furthermore, the development of adaptive neurostimulation devices, capable of the real-time modulation of stimulation parameters based on patient-specific electrophysiological feedback, has opened new avenues for customized rehabilitation strategies that maximize motor circuit activation and recovery potential [57].

While these advancements significantly enhance stem cell-based repair strategies, immune rejection remains a formidable barrier to the clinical translation of biomaterial-assisted stem cell therapies, particularly in allogeneic transplantation settings [59,60]. Advances in CRISPR-based hypoimmunogenic engineering have enabled the generation of HLA-null iPSCs, which evade host immune surveillance while retaining regenerative potential [61]. By selectively deleting classical MHC class I and II molecules while preserving non-classical HLA-G, these engineered stem cells achieve immune evasion without compromising differentiation and functional capabilities [61,62]. Additionally, biomaterial scaffolds engineered to deliver localized immunomodulatory agents, such as interleukin-10 (IL-10) and transforming growth factor-beta (TGF-β), have been shown to create immune-privileged niches, preventing T-cell and NK cell-mediated graft rejection while enhancing transplanted cell survival [63]. Unlike systemic immunosuppression, which carries significant risks of infection, metabolic complications, and immune dysregulation, biomaterial-based localized immunosuppression minimizes adverse effects while optimizing host–graft interactions [59]. Preclinical models have shown that localized IL-10 delivery significantly improves stem cell survival and functional outcomes in SCI, providing a promising immunomodulatory strategy [64]. The ability to customize biomaterials for patient-specific needs, combined with transcriptomic insights into injury-specific microenvironments, could enable highly individualized regenerative strategies that optimize cellular engraftment, circuit reconstruction, and functional recovery.

## 4. Patient-Specific Variables Impacting Stem Cell Therapy for SCI

Another critical determinant of stem cell therapy outcomes in SCI repair is patient-specific variability, encompassing age-related cellular senescence, genetic and epigenetic diversity, injury microenvironmental constraints, and systemic comorbidities [47,65,66,67]. These factors not only modulate the host response to transplantation but also govern the long-term integration of grafted cells into existing neural circuits, ultimately influencing therapeutic efficacy (Figure 3) [65,68]. Aging profoundly alters the cellular and systemic landscape of SCI, introducing biochemical and molecular factors that disrupt neuroregeneration and compromise stem cell function [69]. Hallmarks of age-related dysfunction, including telomere attrition, mitochondrial impairment, and chronic low-grade inflammation (inflammaging), create a hostile post-injury microenvironment characterized by elevated oxidative stress and sustained neuroinflammatory signaling. Increased reactive oxygen species (ROS) accumulation exacerbates neuronal loss, while heightened levels of tumor necrosis factor-alpha (TNF-α) and interleukin-6 (IL-6) intensify glial scarring, further restricting axonal regrowth [66,69,70]. These factors collectively diminish the regenerative potential of stem cells, including NSPCs and iPSCs, exhibiting declined proliferative capacity, impaired lineage commitment, and epigenetic silencing of neurogenic genes with age [71]. Addressing these deficits requires preconditioning strategies, such as antioxidant-based interventions such as N-acetylcysteine to mitigate oxidative stress or the transient expression of Yamanaka factors to rejuvenate aged NSPCs, restoring their proliferative and neurogenic potential [72]. Recent studies had demonstrated that transient Yamanaka factors (*Oct4*, *Sox2*, *Klf4*, and *c-Myc*) expression enhances neuronal differentiation and synaptic integration in aged iPSC-derived progenitors, highlighting its potential for improving SCI repair in older patients [73,74]. As SCI incidence continues to rise in aging populations, the development of age-adapted regenerative protocols is becoming increasingly critical for maximizing stem cell therapy outcomes [1,5].

Additionally, genetic and epigenetic variability further dictates the regenerative potential of transplanted cells, influencing molecular pathways governing neurorepair [71]. Polymorphisms within key signaling cascades, such as *Wnt/β-catenin* and Sonic Hedgehog (SHH), modulate neural differentiation efficiency, while VEGFA variants regulate angiogenesis and graft vascularization [42,71,75]. Similarly, epigenetic modifications, such as DNA methylation and histone acetylation serve as transcriptional regulators of neurogenic genes, with hypermethylation of *NEUROD1* and *ASCL1* linked to impaired neuronal differentiation [71,75]. A recent study using scRNA-seq in rats revealed distinct inflammatory and neurogenic subpopulations that correlate with differential recovery trajectories, highlighting the necessity of patient-specific interventions [76]. The injury microenvironment is another key determinant of stem cell therapy outcomes, as the anatomical level, severity, and chronicity of SCI create biophysical and biochemical barriers to neurorepair [35,77]. Cervical injuries, which disrupt extensive descending motor and autonomic pathways, present greater regenerative challenges than thoracic or lumbar injuries, where residual circuitry is more intact [13]. Severe SCI cases are characterized by widespread neuronal loss, demyelination, and glial scarring, all of which contribute to a neuroinhibitory environment enriched in chondroitin sulfate proteoglycans (CSPGs) that restrict axonal regrowth and limit stem cell integration [23]. A recent preclinical trial demonstrated that region-specific NSPCs, derived based on spatial transcriptomics data, exhibited significantly improved survival and myelination in a rodent SCI model [77]. Further compounding the complexity of SCI repair, systemic comorbidities such as diabetes, obesity, and cardiovascular disease could exacerbate SCI pathophysiology by amplifying chronic inflammation, oxidative stress, and vascular dysfunction; however, limited research has been conducted on the precise mechanisms underlying these interactions and their impact on regenerative therapies [78,79]. Hyperglycemia in diabetic patients has been shown to induce chronic inflammation, impair endothelial function, and restrict blood supply to the injury site, compromising stem cell engraftment and survival [80]. Similarly, obesity-driven metabolic dysregulation disrupts the secretion of neuroprotective factors such as brain-derived neurotrophic factor (BDNF) and insulin-like growth factor 1 (IGF-1), reducing neuronal viability and limiting neurorepair capacity [81,82]. A clinical study evaluating MSC transplantation in diabetic SCI patients reported lower graft survival and integration rates compared to non-diabetic cohorts, highlighting the need for metabolic intervention strategies alongside cell therapy [82]. These findings highlight the critical need for metabolic regulation in conjunction with stem cell therapy, reinforcing the necessity of a precision-medicine approach that accounts for patient-specific physiological and molecular factors to enhance therapeutic efficacy.

## 5. Transcriptomics as a Gateway to Precision Medicine for SCI

Single-cell RNA sequencing (scRNA-seq), spatial transcriptomics, and epigenomic profiling provide an in-depth characterization of injury and patient-specific molecular landscapes, enabling precision-driven approaches to stem cell differentiation, immune evasion, and functional restoration [16,17,77,83]. The resolution afforded by single-cell transcriptomics has illuminated the cellular complexity of SCI, uncovering distinct astrocytic, microglial, and neuronal subpopulations that orchestrate injury progression and repair (Figure 4). Recent scRNA-seq studies have revealed functionally divergent astrocyte phenotypes, including A1 astrocytes, which drive neuroinflammation through IL-1β and TNF-α secretion, and A2 astrocytes, which promote neuroprotection and axonal regeneration via BDNF and GDNF release [84,85]. Similarly, microglial heterogeneity has been dissected at the transcriptional level, demonstrating that M1 microglia perpetuate inflammatory damage, whereas M2 microglia exhibit pro-reparative properties by upregulating *Arg1* and *IL-10* [86]. In addition, scRNA-seq enables the real-time tracking of transplanted stem cells, revealing transcriptional signatures that distinguish successfully integrated neurons and oligodendrocytes from those undergoing apoptosis [16]. Furthermore, scRNA-seq could provide insight into the molecular mechanisms governing lineage specification by identifying how tightly regulated pathways, including Sonic Hedgehog (SHH), Wnt/β-catenin, and Notch, influence the fate of transplanted stem cells, while also revealing how region-specific SHH gradients drive caudalization, facilitating spinal cord-specific oligodendrocyte differentiation, remyelination, and improved axonal conduction [87,88]. Additionally, the spatial transcriptomic mapping of the spinal cord has identified regionally distinct transcriptional programs, facilitating the development of cervical, thoracic, and lumbar NSPC subtypes that are molecularly optimized for site-specific integration [89]. This spatially refined approach extends to growth factor-mediated lineage modulation, as transcriptomic-guided differentiation has identified BDNF and GDNF as key drivers of neuronal maturation, while PDGF-AA supports oligodendrocyte lineage commitment. These insights are now being leveraged in bioengineered hydrogel-based delivery systems, where precisely timed release of trophic factors enhances survival and lineage fidelity of transplanted stem cells [22,64].

The growing application of patient-specific transcriptomic profiling reinforces the shift toward individualized regenerative strategies in SCI. Molecular polymorphisms, such as VEGFA variants influencing angiogenesis and SHH mutations affecting neural differentiation, have been identified as key determinants of stem cell therapy responsiveness [90]. By integrating scRNA-seq with machine learning-based predictive modeling, transcriptomic datasets could now enable patient stratification for clinical trials, optimizing patient selection for iPSC-derived NSPC transplantation based on gene expression biomarkers, including *BDNF*, *NT-3*, and *VEGF* [91]. Additionally, CRISPR-based transcriptomic-guided interventions offer novel avenues for modulating patient-specific gene targets to enhance stem cell resilience [61,92]. Targeted silencing of pro-apoptotic genes and upregulation of neurotrophic factors have demonstrated efficacy in promoting graft survival and functional recovery [14,93]. In a rat model of traumatic brain injury, transplantation of bone marrow MSCs with silenced *Rac1* enhanced survival and improved neurological function by inhibiting NADPH oxidase subunits, thereby reducing oxidative stress and apoptosis [94]. Additionally, the administration of neurotrophic factors has been shown to support neuronal survival and function [95]. While these technologies have refined our understanding of cellular heterogeneity, their application in clinical decision-making remains underdeveloped. Translating transcriptomic insights into personalized therapeutic strategies necessitates an approach that aligns biomarker discovery with real-time patient stratification. Incorporating transcriptomic profiling into clinical workflows would facilitate precision-driven interventions, enabling the selection of patient-specific stem cell populations optimized for engraftment and functional recovery. Moreover, dynamic monitoring of cerebrospinal fluid (CSF) biomarkers, coupled with predictive modeling, could potentially refine immunomodulatory strategies, rehabilitation protocols, and neurostimulation parameters. Future directions could include, establishing a framework that integrates transcriptomics with clinical algorithms, that has the potential to transform SCI treatment paradigms, shifting regenerative medicine toward individualized therapeutic regimens tailored to molecularly defined patient subgroups.

## 6. Conclusions

SCI is a complex, multifactorial condition that cannot be addressed by targeting a single gene or pathway. Unlike monogenic disorders that can be corrected through single-gene modifications, SCI involves a dynamic interplay of injury severity, immune response, extracellular matrix remodeling, and neuroinflammation. Advances in region-specific NSPCs and iPSC-derived neural progenitors have demonstrated promise in preclinical and early-phase clinical studies, with lineage-optimized differentiation enhancing synaptic integration and functional recovery [4,15,18]. Meanwhile, biomaterial-based scaffolds incorporating electrically conductive polymers and growth factor-releasing hydrogels have enhanced cell survival and host integration, accelerating axonal regeneration in preclinical models [21,51,60]. Despite these advances, challenges persist in optimizing immune compatibility, as evidenced by variability in graft survival across patient cohorts [63,68,96]. Strategies such as CRISPR-mediated hypoimmunogenic engineering and localized biomaterial-assisted immune modulation are now emerging as viable solutions for overcoming rejection and improving long-term outcomes [27,61].

Single-cell transcriptomics and machine learning-driven patient stratification are further transforming SCI treatment by enabling personalized regenerative strategies [16,29,91]. Spatially refined transcriptomic profiling has facilitated the development of region-specific NSPCs that optimize site-specific engraftment and functional recovery, with preclinical models demonstrating enhanced myelination and neuronal survival [39,77]. The application of adaptive neurostimulation, in conjunction with stem cell transplantation, has also shown clinical promise, with studies demonstrating improved motor function recovery in SCI patients [8,57]. As these technologies converge, the field is shifting toward precision regenerative medicine, integrating stem cell engineering, immune modulation, bioengineering, and transcriptomics to develop individualized therapies. Among the key stem cell candidates for SCI repair, neural stem/progenitor cells (NSPCs), induced pluripotent stem cells (iPSCs), and mesenchymal stem cells (MSCs) each present distinct advantages and limitations. NSPCs exhibit strong neurogenic capacity and regional specificity, promoting circuit reconstruction and myelination, but face scalability and ethical sourcing challenges [97]. iPSCs offer patient-specific, pluripotent versatility with scalable expansion, though clinical application is hampered by high production costs, tumorigenicity risks, and complex regulatory oversight [38,98]. MSCs, in contrast, are widely accessible and inexpensive to manufacture, exert potent immunomodulatory and trophic effects, but have limited neuronal differentiation capacity and integration potential [44,46]. Clinically, MSCs currently represent the most accessible option, while iPSC- and NSPC-based therapies are advancing in early trials, driven by improvements in manufacturing and immune engineering. Emerging technologies such as CRISPR/Cas9 offer the ability to generate hypoimmunogenic stem cell lines with improved graft survival and reduced rejection, especially for iPSCs and NSPCs [61,92]. Concurrently, machine learning models trained on single-cell transcriptomic data are revolutionizing patient stratification, enabling predictive modeling of treatment response and personalized therapeutic design in SCI repair [28].

In parallel, five core technological platforms—bioengineered scaffolds, growth factor-infused hydrogels, adaptive neurostimulation, transcriptomics, and CRISPR-based gene editing—are catalyzing the next generation of personalized regenerative interventions. Bioengineered scaffolds and 3D-bioprinted matrices provide essential architectural support for stem cell engraftment and axonal regeneration; while materials are scalable and relatively affordable, their customization for individual patient demands advanced fabrication infrastructure and rigorous quality assurance. Growth factor-infused hydrogels are among the most clinically translatable innovations, offering low-cost and modular delivery of neurotrophic signals, although achieving consistent in vivo performance remains a challenge [22,64]. Adaptive neurostimulation devices like epidural electrical stimulation (EES) and functional electrical stimulation (FES) show robust clinical efficacy, but their accessibility is limited by the need for surgical expertise, physical rehabilitation infrastructure, and significant upfront cost [8,58]. Transcriptomic profiling, especially when combined with spatial resolution and machine learning, enables precision-matched stem cell therapies, yet its clinical integration is currently hindered by the cost of high-throughput sequencing, data complexity, and the need for cross-disciplinary workflows [76]. Finally, CRISPR technologies hold transformative potential for creating universal, immune-evasive cell products, but face translational delays due to regulatory and ethical hurdles. For these advanced platforms to be effectively implemented in clinical care, translational strategies must prioritize cost containment, manufacturing automation, cross-sector collaboration, and equitable deployment models, particularly for underserved populations. Synergizing these innovations with stem cell-based therapies is key to realizing scalable, accessible, and individualized solutions for spinal cord injury repair. Future directions will focus on translating these advancements into large-scale clinical applications, ensuring their efficacy and scalability. Translating these advances into clinical solutions will be transformative in restoring function and promoting long-term neurorepair in SCI patients.

## Figures and Tables

**Figure 1 ijms-26-03874-f001:**
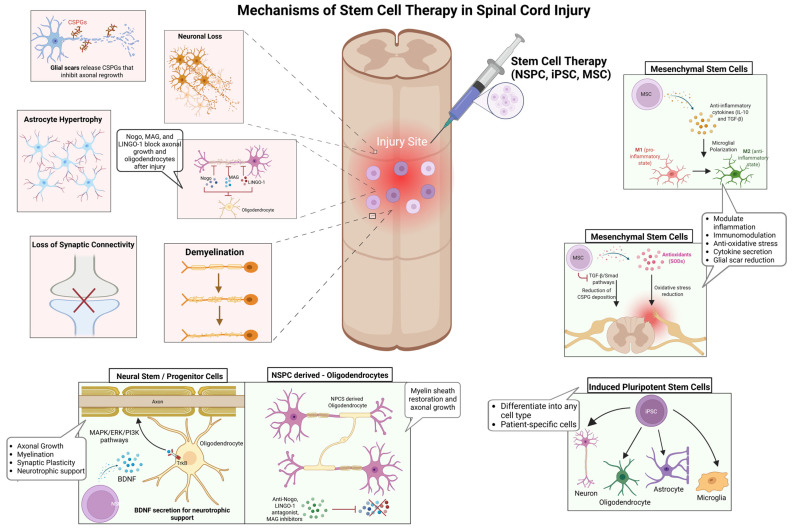
Mechanisms underlying stem cell therapy in SCI repair. SCI disrupts neural circuits through neuronal loss, demyelination, astrocyte hypertrophy, synaptic disconnection, and glial scar formation, creating significant barriers to regeneration. Stem cell-based therapies aim to restore neural function by addressing these pathological hallmarks. NSPCs enhance axonal growth, synaptic plasticity, and remyelination through neurotrophic support. iPSCs generate patient-specific neural derivatives, reducing immune rejection and supporting neural repair. MSCs regulate the injury microenvironment by shifting microglia from a pro-inflammatory (M1) to an anti-inflammatory (M2) state, mitigating oxidative stress, and secreting neuroprotective cytokines such as IL-10 and TGF-β. By targeting key cellular and molecular deficits, these regenerative strategies offer a multifaceted approach to promoting neuroprotection, remyelination, and functional recovery in SCI. Figure created using BioRender.com (accessed 16 April 2025): https://BioRender.com/411iuf7.

**Figure 2 ijms-26-03874-f002:**
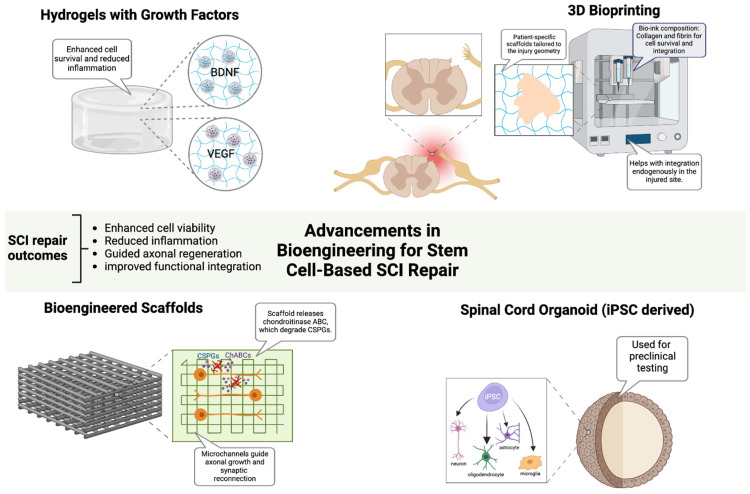
Advancements in Bioengineering for Stem Cell-Based SCI Repair. Bioengineering innovations enhance stem cell-based therapies for SCI repair by improving cell survival, reducing inflammation, and guiding axonal regeneration. Growth factor-infused hydrogels support neuronal viability, while bioengineered scaffolds with aligned microchannels and ChABC promote axonal growth and synaptic reconnection. Three-dimensional bioprinting enables patient-specific scaffold fabrication to optimize cellular integration. iPSC-derived spinal cord organoids recapitulate native spinal architecture, serving as preclinical models for personalized therapy development. Figure created using BioRender.com (accessed 16 April 2025): https://BioRender.com/jupnwy6.

**Figure 3 ijms-26-03874-f003:**
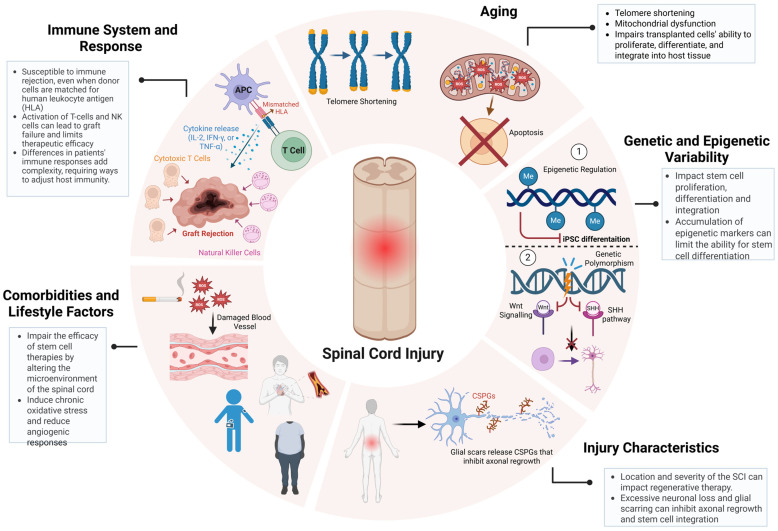
Patient-specific variables impacting the efficacy of stem cell-based therapies for SCI. Key patient-specific factors influence SCI treatment outcomes. Aging-driven telomere shortening and mitochondrial dysfunction impair stem cell proliferation and differentiation. Genetic and epigenetic variability modulate neural integration and regenerative potential. Injury severity, glial scarring, and chronic inflammation create barriers to axonal regrowth. Comorbidities such as diabetes and obesity exacerbate oxidative stress and inflammation, further limiting therapeutic efficacy. Immune rejection by T-cells and NK cells remains a major challenge. Addressing these factors through precision regenerative strategies is crucial for optimizing SCI repair. Figure created using BioRender.com (accessed 16 April 2025): https://BioRender.com/wcuh6xo.

**Figure 4 ijms-26-03874-f004:**
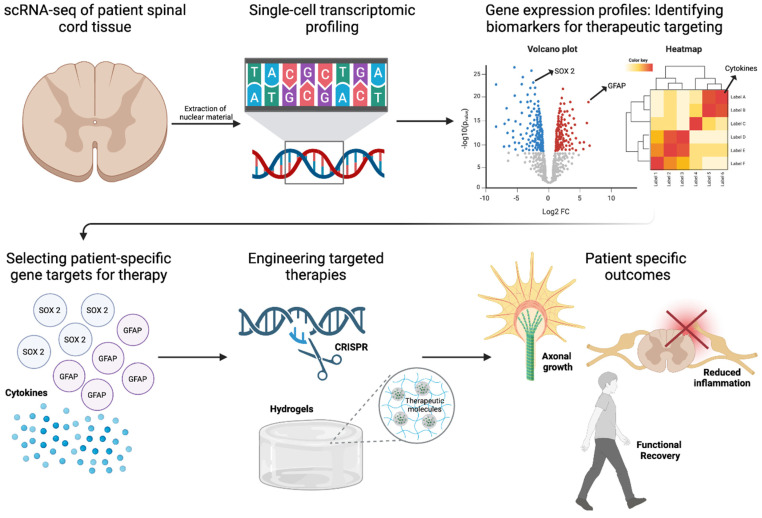
Role of Transcriptomics in SCI Repair. Single-cell RNA sequencing (scRNA-seq) of patient spinal cord tissue enables transcriptional profiling to identify biomarkers for therapeutic targeting. Differential gene expression analysis highlights key regulators such as *SOX2* and *GFAP*, guiding the selection of patient-specific gene targets. Red and blue dots represent significantly upregulated and downregulated genes, respectively, identified via single-cell RNA sequencing. Engineered therapies, including CRISPR-based gene editing and hydrogel-based delivery of therapeutic molecules, are designed to promote axonal growth and reduce inflammation. These precision approaches aim to enhance functional recovery in SCI repair. Figure created using BioRender.com (accessed 16 April 2025): https://BioRender.com/74iqv1j.

## Data Availability

The original contributions presented in this study are included in the article.

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
