# Peer review of "Personalized Stem Cell-Based Regeneration in Spinal Cord Injury Care"

_ijms, 2025, doi:10.3390/ijms26083874_

Round 1

Reviewer 1 Report

Comments and Suggestions for Authors

This manuscript is well-written and well-organized. I suggest only minor revisions on the following points:

  1. It is evident that the authors have used Biorender to create several figures in this manuscript. According to BioRender's user agreement, the software must be properly credited. Additionally, the authors should ensure they have obtained the necessary copyright permissions for publication to avoid any potential legal issues in the future.
  2. The manuscript would benefit from more discussion on how these advancements could be implemented in clinical settings in the future, particularly in terms of cost, scalability, and accessibility. Including these aspects could enhance the paper’s impact and attract broader interest.
  3. The authors should expand their discussion by providing a comparative analysis of different stem cell types (NSPCs, iPSCs, MSCs), highlighting their respective advantages and limitations. Additionally, discussing how emerging technologies such as machine learning and CRISPR could further revolutionize SCI therapies would add valuable depth to the manuscript.

Author Response

Response to Reviewers’ Comments

We would like to start by thanking the comments by the anonymous Reviewer number #1. We are grateful of your time and effort. Your comments have significantly improved the quality of our revised manuscript.   

Reviewer #1:

1) It is evident that the authors have used Biorender to create several figures in this manuscript. According to BioRender's user agreement, the software must be properly credited. Additionally, the authors should ensure they have obtained the necessary copyright permissions for publication to avoid any potential legal issues in the future.

Our response: We thank the reviewer for this important observation. As recommended, we have now included the appropriate credit line with the unique publication license url for each figure. Eg: “Created in BioRender. Lab, T. (2025) https://BioRender.com/411iuf7)” — at the end of each relevant figure legend. Additionally, all figures were created under a paid academic subscription held by the first author, which permits use for journal publications.

2) The manuscript would benefit from more discussion on how these advancements could be implemented in clinical settings in the future, particularly in terms of cost, scalability, and accessibility. Including these aspects could enhance the paper’s impact and attract broader interest.

Our response: We appreciate this valuable suggestion. We have now expanded the discussion to address key translational considerations, including cost-effectiveness, scalability of manufacturing stem cell products, and strategies for enhancing accessibility in diverse healthcare systems. These additions can be found in the revised Discussion section.

3) The authors should expand their discussion by providing a comparative analysis of different stem cell types (NSPCs, iPSCs, MSCs), highlighting their respective advantages and limitations. Additionally, discussing how emerging technologies such as machine learning and CRISPR could further revolutionize SCI therapies would add valuable depth to the manuscript.

Our response: Thank you for this insightful recommendation. In response, we have expanded our discussion to provide a comparative overview of NSPCs, iPSCs, and MSCs, outlining their distinct biological properties, therapeutic potential, and limitations in the context of spinal cord injury repair.

Reviewer 2 Report

Comments and Suggestions for Authors

Thank you for the opportunity to evaluate this manuscript. The text is within the journal scope and the article is interesting.

This review is comprehensive and effectively integrates current advancements in stem cell biology, bioengineering, immunomodulation, and transcriptomics to present a compelling case for personalized regenerative strategies in spinal cord injury care. The manuscript thoroughly covers key cellular therapies (NSPCs, iPSCs, MSCs), their mechanisms of action, and associated challenges in SCI treatment. The detailed discussion of patient-specific variables (e.g., aging, genetic/epigenetic diversity, comorbidities) is particularly commendable.

The manuscript is well-structured, clearly written, and provides a balanced overview of the challenges and future directions in the field. The inclusion of cutting-edge topics such as CRISPR-mediated hypoimmunogenic engineering, 3D bioprinting, and machine learning for patient stratification reflects a strong grasp of the rapidly evolving landscape in SCI therapeutics.

I recommend acceptance in the present form. The manuscript is of high scientific quality and makes a significant contribution to the field.

Author Response

Response to Reviewer 2

Comment 1: Thank you for the opportunity to evaluate this manuscript. The text is within the journal scope and the article is interesting.This review is comprehensive and effectively integrates current advancements in stem cell biology, bioengineering, immunomodulation, and transcriptomics to present a compelling case for personalized regenerative strategies in spinal cord injury care. The manuscript thoroughly covers key cellular therapies (NSPCs, iPSCs, MSCs), their mechanisms of action, and associated challenges in SCI treatment. The detailed discussion of patient-specific variables (e.g., aging, genetic/epigenetic diversity, comorbidities) is particularly commendable. The manuscript is well-structured, clearly written, and provides a balanced overview of the challenges and future directions in the field. The inclusion of cutting-edge topics such as CRISPR-mediated hypoimmunogenic engineering, 3D bioprinting, and machine learning for patient stratification reflects a strong grasp of the rapidly evolving landscape in SCI therapeutics. I recommend acceptance in the present form. The manuscript is of high scientific quality and makes a significant contribution to the field.

Our Response: We sincerely thank the reviewer for taking the time to read and provide such generous feedback on our manuscript. We are grateful for your thoughtful comments regarding the organization, clarity, and scientific value of the review. Your recognition of the integration of emerging technologies; such as CRISPR, 3D bioprinting, and machine learning, as well as the emphasis on patient-specific factors in spinal cord injury care, is deeply appreciated. We are encouraged by your recommendation for acceptance and truly value your support and kind words about the manuscript’s contribution to the field. Thank you!
